# Amino Acid Metabolism and Autophagy in Atherosclerotic Cardiovascular Disease

**DOI:** 10.3390/biom14121557

**Published:** 2024-12-06

**Authors:** Yuting Wu, Irem Avcilar-Kücükgöze, Donato Santovito, Dorothee Atzler

**Affiliations:** 1Institute for Cardiovascular Prevention, Ludwig-Maximilians-Universität München, 80336 Munich, Germany; yuting.wu@med.uni-muenchen.de (Y.W.); irem.avcilar@med.uni-muenchen.de (I.A.-K.); 2DZHK (German Center for Cardiovascular Research), Partner Site Munich Heart Alliance, 80336 Munich, Germany; 3Institute for Genetic and Biomedical Research (IRGB), Unit of Milan, National Research Council, 20133 Milan, Italy; 4Walter Straub Institute of Pharmacology and Toxicology, Ludwig-Maximilians-Universität München, 80336 Munich, Germany

**Keywords:** atherosclerosis, amino acids, autophagy, cardiovascular disease

## Abstract

Cardiovascular disease is the most common cause of mortality globally, accounting for approximately one out of three deaths. The main underlying pathology is atherosclerosis, a dyslipidemia-driven, chronic inflammatory disease. The interplay between immune cells and non-immune cells is of great importance in the complex process of atherogenesis. During atheroprogression, intracellular metabolic pathways, such as amino acid metabolism, are master switches of immune cell function. Autophagy, an important stress survival mechanism involved in maintaining (immune) cell homeostasis, is crucial during the development of atherosclerosis and is strongly regulated by the availability of amino acids. In this review, we focus on the interplay between amino acids, especially L-leucine, L-arginine, and L-glutamine, and autophagy during atherosclerosis development and progression, highlighting potential therapeutic perspectives.

## 1. Introduction

As the leading cause of mortality worldwide, it is reported that cardiovascular disease (CVD) accounts for 45% of all deaths in Europe [1]. It causes over 3.8 million deaths annually in Europe [2], and its estimated healthcare cost is more than EUR 200 billion per year [1]. Atherosclerosis is the key driver of CVD and has been associated with elevated plasma cholesterol levels, as revealed by multiple epidemiological cohorts and confirmed by randomized clinical trials with lipid-lowering treatments. However, other risk factors, such as smoking, aging, and physical inactivity, also trigger its prevalence and the incidence of acute CV events. Current therapies aim to prevent CVD primarily by targeting blood lipid levels, such as statins or PCSK (proprotein convertase subtilisin/kexin type) 9 inhibitors. However, such lipid-lowering treatments are not effective in lowering the CV risk in all patients and a considerable residual risk of developing CVD persists despite the optimal medical management of plasma lipid levels. Recently, the collaborative analysis of the PROMINENT (NCT03071692), REDUCE-IT (NCT01492361), and STRENGTH (NCT02104817) trials revealed that inflammation is an even stronger predictor of risk for future CV events or death than plasma cholesterol levels in patients receiving statins [3]. Likewise, Bay et al. observed an independent association between major adverse CV events (MACE) and high-sensitivity C-reactive protein, a biomarker of subclinical systemic inflammation, in patients with polyvascular atherosclerotic disease [4]. The inflammatory hypothesis of atherosclerosis has been validated in various clinical trials, pointing out that specific anti-inflammatory therapeutics (e.g., canakinumab or colchicine) are able to reduce MACE incidence [5,6,7]. However, other anti-inflammatory treatments, such as methotrexate, varespladib, or darapladib failed to show clinical benefit in patients with CAD [8,9,10]. Therefore, novel tailored therapeutics specifically targeting atherosclerotic inflammation without compromising the host’s immune defense are urgently needed.

During atherogenesis, the interplay between various innate and adaptive immune cells, but also non-immune cells, is a key driver for disease progression. The altered metabolism of immune and non-immune cells is of great importance to cell function and the subsequent atherosclerosis development. In this process, amino acids (AAs) and their metabolism are critical regulators in numerous pathways related to atherogenesis. In recent years, autophagy emerged as an important regulator of cell adaptation to stress with important implications for cell homeostasis and metabolism. Autophagy is an evolutionarily conserved catabolic mechanism which is implicated in the removal of damaged and/or dysfunctional intracellular components and plays important roles in the development and progression of atherosclerosis. The AAs L-arginine (Arg), L-glutamine (Gln), and L-leucine (Leu) have been identified as modulators in the context of both atherogenesis and autophagy. In this review, we summarize the current understanding of the relationship between autophagy and atherosclerosis and the role of AAs in these processes.

## 2. Pathogenesis of Atherosclerosis

Atherosclerosis is a chronic inflammatory disease affecting large- or medium-sized arteries, characterized by the buildup of plaques that limit blood flow, eventually reducing tissue perfusion (Figure 1). This process is initiated by endothelial dysfunction, followed by the infiltration of low-density lipoprotein (LDL) and immune cells into the intima, leading to the formation of fatty streaks. As a result, lipid accumulation occurs, the plaque enlarges, and fibrous elements form over the plaque, causing vessel narrowing and activating inflammatory pathways (Figure 1) [11]. Endothelial cells (ECs) cover the inner surface of all blood vessels. Therefore, they directly sense mechanical forces from blood flow. In areas of curved or bifurcated arteries, the blood flow pattern is dominated by oscillating shear stress (OSS), which directly affects the morphology and function of the endothelium [12,13]. When exposed to this OSS environment, ECs are transformed into pro-inflammatory, pro-oxidative phenotypes with metabolic abnormalities, mitochondrial dysfunction, and increased permeability. This results in higher LDL retention and oxidation in the intima. Oxidized LDL (oxLDL) stimulates ECs to express adhesion molecules (e.g., P-selectin, vascular cell adhesion molecule-1 (VCAM-1), and intercellular adhesion molecule-1 (ICAM-1)), leading to monocyte, mast cell, and T cell recruitment in the intima, where monocytes differentiate into macrophages. Along with vascular smooth muscle cells (VSMCs), macrophages take up oxLDL to form foam cells [14,15].

In addition to foam cell formation, macrophages contribute to plaque formation through their differentiation into classically activated macrophages (CAMs) and alternatively activated macrophages (AAMs). CAMs are pro-inflammatory, contributing to the progression of atherosclerotic plaques. Conversely, AAM are considered anti-inflammatory with plaque-stabilizing effects due to their capability of clearing lipid debris and apoptotic foamy cells [16]. This overall inflammatory environment leads to the formation of fatty streaks, which is considered the first stage of plaque formation. This process is characterized by a substantial buildup of lipids inside the cells and the extracellular space.

Besides innate immune cells, T cells play a pivotal role during atherogenesis in shaping the inflammatory environment and contributing to plaque development and stability. In the intima, T cells secrete inflammatory cytokines, such as TNF-β, IFN-γ, and lymphotoxin, that induce VSMC migration from media to intima [17]. VSMCs undergo phenotypic switching from a differentiated form to a synthetic form, becoming more proliferative and less contractile [18]. Synthetic VSMCs proliferate and synthesize extracellular matrix proteins, e.g., collagen, elastin, and proteoglycans, which contribute to the development of neointima in an early phase of atherosclerosis and in the formation of a fibrous cap over the lipid-rich core in more advanced phases. Within advanced plaque, the necrotic core contains dead or dying foam cells, lipids, and cellular debris. The necrotic core and the fibrous cap are the hallmarks of advanced atherosclerosis, and atheroma plaque regression is not anticipated in this stage. Due to impaired efferocytosis and macrophage death, the necrotic core further expands. This continuous accumulation can lead to plaque instability and increases the risk of plaque rupture, potentially causing myocardial infarction or acute ischemic stroke.

## 3. Autophagy in Atherosclerosis

Autophagy is a ubiquitous cellular mechanism that allows the orderly digestion of cell components. Autophagy generally acts as a housekeeping mechanism and occurs at low levels under physiological conditions. When cells are stimulated by environmental stress, such as oxidative stress, nutrient deficiency, or hypoxia, autophagy is activated as an adaptive response, which promotes cell survival by recycling precursors (e.g., AAs and free fatty acids) derived from the degradation of endogenous cellular components. Autophagy is characterized by the formation of double-membrane vesicles called autophagosomes, which engulf the intracellular cargo and target lysosomes. When autophagosomes fuse with lysosomes, a series of hydrolases digest the contents of the autolysosomes, which are then released to the cytosol for recycling or reuse [19]. Based on its physiological function and the delivery route of the cargo to the lysosomes, autophagy has been identified as macroautophagy, molecular chaperone-mediated autophagy, and microautophagy [20]. Additionally, autophagy can process bulk or selective cargoes. Bulk autophagy occurs typically during macroautophagy in response to nutrient starvation. In contrast, selective autophagy targets specific cargo for degradation (e.g., lipophagy targets lipid droplets for degradation) and involves molecular adaptors to convey selective cargoes to autophagosomes [21].

Studies over the past decade have demonstrated the importance of autophagy in the development and progression of atherosclerosis and plaque instability [22,23,24]. Perrotta detected autophagy in ECs, macrophages, and VSMCs in human atherosclerotic plaques using transmission electron microscopy [25]. Autophagic cells are mainly located in the fibrous cap and near the necrotic core [25]. The expression of microtubule-associated protein light chain 3 (LC3)-II, a marker of autophagy, is significantly increased in human carotid plaques compared with carotid artery samples from healthy individuals [26]. Autophagy has a dual role in the progression of atherosclerosis [27]. Oxidative stress is a hallmark of advanced atherosclerotic plaques. Mild oxidative stress activates autophagy to facilitate the removal of damaged organelles, contributing to cellular recovery. If autophagy is not sufficient to clear cellular damage (e.g., in the case of severe oxidative stress), cells may undergo apoptosis by activating the caspase cascade [28]. Besides its important anti-apoptotic role in atherosclerotic plaques, autophagy also downregulates apolipoprotein B (ApoB)-containing lipoproteins in the circulation, which would prevent atherogenesis [29]. Despite its protective role, autophagy is responsible for the formation of ceroid, which is an insoluble complex of protein associated with oxidized lipids found in the lesions. Many cells in advanced human plaque contain a large number of lysosomes containing ceroid and a variety of lysosomal enzymes that degrade them [30]. These lysosomal enzymes lose their useful functions, leading to impaired autophagy and induction of apoptosis. This duality underscores the context-dependent nature of autophagy in atherosclerosis.

Autophagy features cell-specific contributions in the maintenance of vascular integrity and health, involving multiple functions in stromal and immune cells. Autophagy contributes to the ability of ECs to adapt to physiological shear stress [31]. ECs exposed to low shear stress have defective autophagic flux in both humans and mice [32]. It has been demonstrated that the impairment of the autophagic machinery in ECs increases the levels of von Willebrand Factor (vWF), P-selectin, VCAM-1, and ICAM-1, which accelerates the development of atherosclerosis [33,34]. Additionally, defective autophagy increases endothelial apoptosis, thus decreasing endothelial integrity and promoting the development of atherosclerosis [32,35]. However, excessive autophagy in ECs may also lead to plaque instability by inducing cell death [23]. Furthermore, autophagy is essential in VSMC phenotypic changes and plasticity. Platelet-derived growth factor (PDGF), which is well known to stimulate VSMC proliferation and migration, induced VSMC autophagy as determined by detection of LC3-II expression and observation of autophagosomes [36]. Autophagy activation by PDGF enhanced the degradation of contractile proteins, thereby allowing VSMCs to reach the synthetic state rapidly [36]. These results point to the crucial role of autophagy in both EC and VSMC function in atheroprogression.

Autophagy is the key to maintaining immune cell function. Autophagy activation protects monocytes from apoptosis and prolongs cell viability by inhibiting caspase 8 [37]. Zhao et al. found that autophagy of circulating monocytes attenuates the vulnerability of coronary atherosclerotic plaques [38]. Furthermore, autophagy contributes to recruitment and differentiation of monocytes. Macrophage-colony-stimulating factor (M-CSF) promotes autophagy by enhancing the phosphorylation of Unc-51-like kinase 1 (ULK1), which promotes the differentiation of monocytes into macrophages [39,40]. In foam cells, autophagy involves the delivery of oxLDL to lysosomes for hydrolysis, which is beneficial for reversing of normal lipid homeostasis [24,41]. Studies have demonstrated that macrophage autophagic dysfunction occurs in both murine and human atherosclerotic lesions, which are mainly characterized by the accumulation of autophagosomes and impaired autophagic flux, and defective fusion of autophagosomes with lysosomes [42,43]. Deletion of key autophagy-related genes (e.g., Atg5, Atg7, Atg14) in mouse macrophages accelerates the development of atherosclerosis as evidenced by increased apoptosis and necrotic cores [43]. Moreover, autophagy has been reported to regulate macrophage polarization towards an anti-inflammatory phenotype through NF-κB degradation and SIRT1 and mTOR pathways [44,45,46]. Altogether, these findings suggest that autophagy contributes to monocytes recruitment and differentiation and promotes cholesterol efflux and anti-inflammatory macrophage polarization.

Autophagy plays a crucial role in immune responses. Studies have demonstrated that autophagy is induced in CD4^+^ T cells upon activation. Li et al. detected autophagosomes in in vitro activated CD4^+^ T cells [47]. They also determined that T cell receptor (TCR) signaling could sustain autophagy in effector CD4^+^ T cells. Atg5-deficient T cells exhibited impaired proliferation after anti-CD3 stimulation but normal TCR expression levels [48]. These findings were confirmed in other mouse models of Atg3, Atg5, Atg7, and Vps34 deletion in CD4^+^ T cells [49,50,51]. Limited studies have also suggested a role for T cell autophagy in atherosclerosis. Amersfoort et al. found that deficiency of T cell autophagy hampered atherosclerosis using T-cell-specific Atg7-deficient (LckCreAtg7fl/fl) mice [52]. Mandatori et al. indicated that impaired autophagy in Tregs is associated with atherosclerosis [53]. These results underline the important role of autophagy in T cell activation and development in atherosclerosis.

In summary, autophagy modulates immune cell and non-immune cell function in various ways, including monocyte differentiation, immune response, and VSMC phenotype changes. Its complex role highlights the importance of autophagy in maintaining cellular homeostasis in cardiovascular health.

## 4. Amino Acid Metabolism in Atherosclerosis

Beyond building blocks of proteins, AAs and their metabolism play important roles in a variety of cellular functions, including energy metabolism, immune responses, and the regulation of oxidative stress. Specific AAs and their metabolites can modulate inflammatory pathways, EC function, and autophagy, all of which are critical in the pathogenesis of atherosclerosis. Understanding the function of AA metabolism is crucial to provide new approaches for targeted therapies in atherosclerosis. To date, the AAs Arg, Leu, and Gln have been reported to functionally regulate autophagy [54] and play a vital role in the progression of atherosclerosis. In this review, we therefore focus on the emerging roles of Arg, its derivative homoarginine (hArg), Gln, and Leu and their metabolisms in the key characteristics of atherosclerosis.

### 4.1. L-Arginine

The importance of Arg metabolism has been well established in both normal and pathological states of the CV system. Arg is a proteinogenic amino acid synthesized from citrulline in the urea cycle. It is metabolized through three primary pathways: In the first pathway, utilized by NO synthase (NOS), Arg is converted to citrulline, producing nitric oxide (NO) as a by-product. In mammals, there are three NOS isoforms: eNOS, endothelial NOS; nNOS, neuronal NOS; and iNOS, inducible NOS, which is primarily expressed in immune cells, such as CAM during infection or inflammation [55]. The second pathway, mediated by arginase, breaks Arg down into ornithine and urea. In mammals, there are two isoforms of arginase: ARG1, which is located in the cytosol, and ARG2, mainly expressed in mitochondria. ARG1 is highly expressed in AAM, and used as an AAM marker, whereas ARG2 may have pro-inflammatory properties and promotes a CAM phenotype [56]. In the third pathway, L-arginine–glycine amidinotransferase (AGAT) transfers an amidino group from Arg to glycine to produce ornithine and guanidinoacetate (GAA), the precursor of creatine. This pathway is the first and rate-limiting step in creatine biosynthesis, pointing to Arg’s role in creatine production and energy metabolism.

As described above, endothelial NO is required for healthy vascular function [55]; therefore, Arg metabolism becomes particularly important for maintaining vascular health and preventing atherosclerosis. Researchers attempted to increase NO bioavailability in order to improve CV outcomes. In animal models of CVD, acute Arg administration has been shown to improve endothelial function [57,58,59,60]. Clinical studies indicate that short-term Arg administration enhances endothelial function in patients with hypercholesterolemia [61], myocardial infarction [62], and critical limb ischemia [63]. Nevertheless, conflicting findings from other clinical studies suggest that Arg supplementation does not affect endothelial function in healthy adults [64] and fails to alleviate CV conditions in patients with hypertension [65] and coronary artery disease [66]. Even worse, Arg administration has been associated with increased mortality when used with standard postinfarction therapies [67]. One possibility is that the expression and activity of arginases increase with exogenous Arg supplementation [68]. Therefore, the excessive amount of Arg is metabolized via arginases, resulting in impaired NO production [68]. These complexities clearly indicate that further studies are required to develop tailored cell-specific Arg-based therapies to improve CV health.

Other studies have highlighted the distinct and complex activities of ARG1 and ARG2 in vascular health. The former plays a protective role, while ARG2 is associated with vascular dysfunction and atherosclerosis progression [69]. The study noted that endothelial ARG activity was increased in mice fed a high-cholesterol diet, resulting in impaired vascular function [70]. Deficiency of Arg2 has been shown to restore vascular NO production, improve endothelial function and reduce atherosclerotic lesions in *Apoe**^-/-^* mice [70,71]. In advanced stages of human carotid plaques, the activation of ARG2 in ECs and macrophages results in increased ROS production [72]. Unlike ARG2, elevated levels of ARG1 have been shown to reduce atherosclerotic features, such as decreased inflammatory cytokines and stimulated VSMC proliferation [73]. Of note, inhibition of both ARG1 and ARG2 promotes endothelial function through NO signaling in *Apoe^-/-^* mice [74]. Furthermore, a recent study found that *Arg1* deficiency in murine erythrocytes increases vascular NO bioactivity and promotes VSMC osteoblastic differentiation and atherosclerotic lesion calcification [75]. Our group recently demonstrated an accumulation of Arg1^+^ lesional macrophages as well as Arg1^+^ lesional T cells in advanced plaques as compared to early lesions in Apoe-deficient mice, suggesting ARG1 contribution to both myeloid and lymphoid cell regulation during atherogenesis [76]. In clinical trials, the ARG inhibitor nor-NOHA has been found to improve endothelial function in elderly healthy subjects [77] and patients with coronary artery disease and type 2 diabetes mellitus [78]. Although these findings are promising, they still emphasize the complex roles of ARG1 and ARG2 in vascular health and indicate the requirement of isoform-specific and cell-specific ARG inhibitors for more targeted treatments.

### 4.2. L-Homoarginine

hArg is synthesized by AGAT, which catalyzes the transfer of an amidino group from Arg to lysine. Low circulating hArg has been considered as a biomarker for CV disorders [79,80,81]; likewise, high concentrations of hArg are associated with a lower risk of CV outcome and mortality [82]. The Mendelian randomization analysis, however, did not show evidence of causal relationships between plasma hArg and any of the studied cardiometabolic outcomes [83].

Dietary intake of hArg increase plasma levels of hArg [84,85]. Animal studies have shown that dietary hArg supplementation improves heart function in rats [86] and mice [87]. Moreover, hArg supplementation rescues impaired cardiac contractile function in *Agat* knockout mice [88]. In the past, people believed that one of the explanations for the beneficial effects of hArg on CV disorders is that hArg may serve as a substrate for NOS, leading to the generation of NO, although hArg has been found to have a much lower affinity to several NOS isoforms compared to Arg [89,90]. Tommasi et al. proposed that hArg inhibits arginases, resulting in higher Arg availability for NO synthesis; however, this idea has not been validated in vivo yet [91]. Our recent study demonstrated that hArg supplementation reduces atherosclerotic plaque development by modulating CD4^+^ T cells [85]. hArg supplementation affects the T cell cytoskeleton by binding to myosin heavy chain 9, leading to changes in T cell activation, motility, migration, and proliferation [85], suggesting that hArg can modulate immune response through CD4^+^ T cells in atherosclerosis. These results support the need for more research to better understand how hArg modulates the immune system and subsequent CV conditions in humans. Notably, oral hArg supplementation is well tolerated in humans without vascular or neurological abnormalities [92]. A clinical trial is currently ongoing to investigate the effect of hArg supplementation in post-stroke patients (www.clinicaltrials.gov, NCT03692234). Further studies are needed to establish the role of this AA in the pathogenesis of atherosclerosis and delineate the cohorts of patients who would specifically benefit the most.

### 4.3. L-Glutamine

Gln is the most abundant free AA in the human body, comprising about 50% of the total free AA pool in tissues such as liver and muscles [93]. Gln is a conditional essential AA, because under conditions of catabolic stress and critical illness, endogenous synthesis of Gln may be inadequate. Glutaminase (GLS) breaks down Gln into Glu and ammonia (NH_3_). Glu is further converted to α-ketoglutarate, which enters the tricarboxylic acid (TCA) cycle for the generation of ATP or acts as an anaplerotic source of carbon for the synthesis of non-essential AAs and lipids. GLS is a mitochondrial enzyme that exists in two isoforms: GLS1 and GLS2. Jha et al. demonstrated that Gln-related metabolism is critical for macrophage polarization, with one-third of all carbons in the TCA cycle metabolites in AAM being derived from Gln [94]. In the absence of Gln, AMM polarization decreased by about 50%, along with a reduction in the production of C-C motif chemokine ligand 22 (CCL22) [94], pointing the necessity of Gln for macrophage differentiation. Additionally, another study observed enhanced Gln deposition in AMM [95], highlighting the importance of Gln uptake by these macrophages. Gln accumulation is not uniform along the aorta from *Ldlr^-/-^* mice fed a high-fat diet for 3 months; in the distal abdominal aorta, Gln accumulation is more pronounced compared to the proximal common iliac arteries [95]. These findings suggest that Gln metabolism and its spatial distribution in the vasculature may be critical for macrophage polarization and its involvement in atherosclerosis.

Beyond macrophage polarization, Gln metabolism also plays an essential role in EC function. In the absence of Gln or when GLS1 is inhibited, EC proliferation and migration are significantly impaired, leading to defects in vessel sprouting [96,97]. Researchers have demonstrated that GLS1 fuels the TCA cycle, providing essential macromolecules for proliferation and motility [97]. Additionally, NH3, which is the by-product of GLS1, promotes EC survival [98]. Exogenously administered or Gln-derived NH3 induce the expression of heme-oxygenase-1 (HO-1) in ECs through the ROS-Nrf2 pathway [98]. HO-1 degrades heme into bilirubin, carbon monoxide (CO), and iron. Bilirubin acts as a potent scavenger of ROS, while CO induces vasodilation and prevents vascular cell apoptosis [99]. These studies indicate the critical role of Gln in supporting EC function, not only by fueling energy metabolism but also through its metabolites.

Several studies have revealed that plasma Gln levels and the Gln-to-Glu ratio (Gln–Glu) have an inverse relationship with higher body mass index, blood pressure, and insulin resistance [100,101,102,103,104]. Post-ischemic reperfusion of rat hearts with Gln resulted in a complete recovery of cardiac output [105,106]. Gln supplementation alleviated atherosclerosis through downregulation of systemic pro-inflammatory pathways in *Apoe^-/-^* mice fed a high-fat diet for 18 weeks [107]. In small clinical trials, Gln supplementation reduced myocardial injury and clinical complications after coronary revascularization [108,109]. Similarly, a case–cohort study involving over 1,000 participants found that Gln–Glu was related to decreased risk of CVD [110]. Furthermore, a study conducted in two large, well-defined, and independent cohorts revealed that dietary intake of Gln was inversely related to the risk of CV mortality [111]. These findings suggest that Gln and its metabolism could be a valuable therapeutic target for managing atherosclerosis and improving CV outcomes.

### 4.4. L- Leucine

Leu is an essential AA that cannot be synthesized by the human body and has to be taken through the diet. It is a branched-chain AA (BCAA) and most of it primarily contributes to protein synthesis within the skeletal muscle [112]. Approximately 20% of Leu is broken down via branched-chain aminotransferase (BCAT), which catalyzes the transamination reaction to produce α-ketoisocaproate. This compound is further metabolized into the ketone body acetoacetate and acetyl-coenzyme A (acetyl-CoA) [113], which are subsequently oxidized in TCA cycle.

The BCAAs Leu, isoleucine, and valine share similar metabolic pathways due to their chemical similarity and often exhibit synergistic effects on metabolic processes. Consequently, there is a substantial body of research exploring their collective impact on cardiometabolic diseases and their risk factors. Several studies reported that plasma levels of BCAAs are strongly associated with insulin resistance and coronary heart disease [101,114,115,116]. BCAA supplementation induced eNOS expression, ROS production, and pro-inflammatory status in ECs and peripheral blood mononuclear cells, leading to endothelial dysfunction [117,118]. Elevated BCAA promotes atherosclerosis progression in *Apoe*^-/-^ mice fed a high-fat diet through pro-inflammatory macrophages activation [119]. Notably, a study involving over 400 participants demonstrated that circulating BCAA levels are significantly and positively correlated with risk factors of coronary artery disease, such as carotid intima-media thickness, body mass index, waist circumference, blood pressure, and fasting blood glucose [120]. This association may be linked to BCAA catabolic defects [119,121]. He et al. analyzed transcriptomes of the plaques at various stages and revealed that Leu degradation gene set significantly enriched in early-stage plaques compared to advanced-stage lesions [121]. A recent study identified the individual roles of Leu in plasma, revealing that higher blood Leu levels activate mTOR signaling pathway in human and mouse monocytes/macrophages, which in turn promotes atherosclerosis in mice [122].

On the other hand, several studies have shown conflicting results, suggesting that Leu may play a protective role in atherosclerosis. For instance, Leu supplementation decreases macrophage lipid content in macrophages with improved mitochondrial respiration [123,124]. Grajeda-Iglesias et al. also found that serum collected from healthy volunteers after Leu supplementation (5 g/day for 3 weeks) showed anti-atherosclerotic properties [123]. A 1.5% Leu supplementation in drinking water for 8 weeks resulted in approx. 60% reduction in aortic atherosclerotic lesion area in *Apoe*-deficient mice, along with improved hepatic lipid metabolism and reduced systemic inflammation [125]. Similarly, supplementation with BCAA has been associated with a reduction in atherosclerotic lesion area, as well as a significant decrease in serum cholesterol and LDL cholesterol levels [126]. Additionally, BCAA supplementation was found to lower the systemic inflammatory responses and downregulate inflammatory-related signaling pathways [126]. Bcat1 overexpression triggers lipid accumulation in aortic tissues and aggravated plaque inflammation in *Apoe^-/-^* mice fed a high-fat diet [127]. This conflicting evidence suggests that the role of BCAAs, and specifically Leu, in CV health is complex and may depend on various factors. Further research focusing specifically on Leu, rather than BCAAs collectively, might provide clearer insights into its distinct cell-specific effects on CV health.

## 5. The Regulatory Role of Amino Acids on Autophagy

It is well established that autophagy is triggered by starvation, which includes the depletion of AAs (Figure 2). Autophagy could also regulate the abundance of free AAs via recycling intracellular constituents [128]. Two evolutionarily conserved nutritional pathways, the mTORC1 and general control nonderepressible 2 (GCN2) pathways, occupy pivotal positions in AA sensing [129]. AA starvation activates the autophagy signaling pathway through inhibition of mTORC1 signaling and activation of GCN2 signaling, which releases AAs by degrading proteins to maintain the AA pool for important protein synthesis.

### 5.1. Amino Acids Regulate Autophagy via mTORC1 Pathway

The intracellular levels of AAs have a strong impact on the sensitivity of mTORC1. The heterodimeric Rag GTPases, consisting of RagA/B and RagC/D play a pivotal role in regulating mTORC1 by nutrients. In response to AAs, RagA/B in its GTP-bound form and RagC/D in its GDP-bound form facilitate the recruitment of mTORC1 to the surface of lysosomes and enhance its kinase activity (Figure 3) [130]. Activated mTORC1 inhibits autophagy initiation by phosphorylating proteins, such as ULK1, Atg13, Atg14L, and nuclear receptor-binding factor 2 (NRBF2). This phosphorylation prevents the production of phosphatidylinositol 3-phosphate (PI3P), which is necessary for phagophore enucleation and the initiation of autophagy [131,132,133,134]. AA starvation causes the dephosphorylation of ULK1 by inhibiting the activation of mTORC1, which in turn triggers autophagy [135,136]. Wong et al. identified another mechanism of ULK1 activation: AA starvation promotes the dissociation of protein phosphatase 2A (PP2A) from the inhibitory factor alpha4 [137]. The mTORC1 pathway activation is unequally sensitive to different AAs. This review aims at clarifying the function of Arg, Gln, and Leu in the process of mTORC1 activation.

#### 5.1.1. Arginine and Autophagy

CASTOR1, the solute carrier family 38A9 (SLC38A9) and Transmembrane 4 L six family member 5 (TM4SF5) act as ubiquitous Arg sensors for the mTORC1 pathway. Structural analysis has shown that Arg binds to two ACT domains (aspartate kinases, chorismate mutase, and TyrA) of CASTOR1. This binding leads to an allosteric regulation in the area near the GATOR2-binding site. As a result, GATOR2 dissociates from CASTOR1 and activates mTORC1 downstream [138]. SLC38A9 is a lysosomal 11-transmembrane-segment protein, which interacts with the Ragulator-Rag GTPase complex and stimulates the activity of mTORC1 [139,140,141]. Furthermore, SLC38A9 transports several essential AAs including Leu in an Arg-regulated manner and is essential for mTORC1 activation [142,143]. TM4SF5 forms a complex with mTOR and SLC38A9 on lysosomal membranes in an Arg-regulated manner, activating mTOR/S6K1 [144]. The metabolism of Arg produces NO and citrulline, which also play a role in regulating mTORC1 signaling. NO impairs the activity of JNK1 and phosphorylation of Bcl-2, while activating mTORC1, this effect is dependent on the presence of IKKβ and TSC2 [145].

Previous studies have focused on Arg’s influence on the atherosclerotic process through different metabolic pathways (NOS or ARG). It is important to note that Arg could also have various effects on the different stages of atherosclerosis by modulating the mTORC1 autophagy pathway. García-Narvas et al. [146] discovered that the depletion of Arg inhibited the expression of various membrane antigens (including CD3ζ) in human T cells, leading to endoplasmic reticulum stress (ERS). Subsequent triggering of autophagy by Arg depletion keeps T cells alive under ERS conditions. Supplementation of Arg to autophagy-impaired cells with lysosomal insufficiency partially restored mTOR activity and ameliorated cytopathological abnormalities [147], suggesting that autophagy impairment in atherosclerosis may be ameliorated by Arg supplementation. However, novel aspects of the crosstalk between Arg, its metabolism, and autophagy are emerging, and further research is warranted to understand molecular mechanisms and disease relevance. For example, ARG2 has been shown to suppress autophagy in ECs in atherosclerotic mice. Still, the effect was observed in both overexpressing the wild-type and the enzymatically inactive enzyme, suggesting a regulatory role that may be independent of Arg [71]. Additionally, the possible role of hArg in the mechanisms of autophagy activation in atherosclerosis is still to be explored.

#### 5.1.2. Leucine and Autophagy

Previous investigations have stated that Sestrin2, a ubiquitous highly conserved stress-responsive protein functions as a Leu sensor within the mTORC1 pathway [148]. When the levels of Leu increase, Leu attaches to Sestrin2, causing Sestrin2 to separate from GTPase-activating proteins toward Rags 2 (GATOR2). This results in increased activity of GATOR2 and subsequently suppression of GATOR1, followed by the activation of Rag A/B. Rag A/B ultimately triggers the activation of mTORC1 and hinders the process of autophagy [149]. Chen et al. discovered that in the condition of Leu sufficiency, SAR1B attaches to Leu, undergoes a conformational change and dissociates from GATOR2, which results in mTORC1 activation [150]. Wyant et al. identified SLC38A9 as a Leu transporter. Under long-term Leu starvation, SLC38A9 transports the Leu generated through lysosomal proteolysis into the cytosol, where it reactivates mTORC1 through cytosolic mechanisms [143]. In addition to acting directly on the sensor, the Leu metabolite acetyl-CoA also positively controls the activity of mTORC1 by acetylating Raptor [151].

Current studies suggest that the role of Leu in atherosclerosis is controversial. The knowledge on the effect of Leu on autophagy during atherogenesis is still incomplete. Some studies have suggested that Leu supplementation has an anti-atherosclerotic effect [123,125]. However, a recent study by Zhang et al. indicated that the consumption of dietary high protein acutely elevated AA levels (Leu in particular) in the blood and triggers Leu-mediated mTOR activation in human monocytes/macrophages and drives atherosclerosis in mice [122]. On the other hand, it has been found that cytosolic BCAT expression is increased upon T cell activation, which limits Leu availability to mTORC1 signaling [152]. More research is needed to determine the cell-specific effects of Leu on autophagy in the context of atherosclerosis and the molecular mechanisms involved.

#### 5.1.3. Glutamine and Autophagy

Gln is imported via the SLC1A5 transporter [153]. Part of the Gln can be efficiently exported in the exchange of essential AAs via SLC7A5/SLC3A2 transporters. Gln activates mTORC1 by two ways (Figure 3): First, Rag-dependent: the production of α-ketoglutarate from Gln stimulates the activation of Rag GTPase, which recruits mTORC1 to the lysosome where it becomes activated and subsequently inhibits autophagy [154]. Second, Rag-independent and Phospholipase D (PLD)-dependent: Bernfeld et al. [155] demonstrated that PLD-derived phosphatidic acid is necessary for the activation of mTORC1 by Gln, even in the absence of Leu or RagA/B. During Gln deprivation, autophagy-deficient cells experience an increase in the absorption of AAs from the extracellular compartment. This is due to an upregulated gene expression of activating transcription factor 4 (ATF4)-dependent AA transporter [156]. This suggests that Glu is a negative regulator of autophagy.

### 5.2. Amino Acids Regulate Autophagy via GCN2 Pathway

GCN2 is a direct sensor of AA deprivation. During AA starvation, uncharged tRNAs accumulate in the ribosome’s A-site and interact with GCN2, promoting its dimerization and autophosphorylation [156,157]. Activated GCN2 phosphorylates the eukaryotic translation initiator factor 2α (eIF2α) at Ser51, which affects autophagy at the transcriptional level. Phosphorylating eIF2α can increase the expression of ATF4 and C/EBP-homologous protein (CHOP), which in turn promotes autophagy by boosting the transcription of autophagy-related proteins, such as Atg5, Atg7, and Atg10 [158].

The GCN2 and mTORC1 signaling pathways coordinate cellular adaptation to AA deprivation through complementary mechanisms. Activation of GCN2 by AA starvation increases the expression of ATF4 and facilitates the expression of the stress response protein Sestrin2. Sestrin2 is necessary to maintain the regulation of mTORC1 by preventing its localization to the lysosomes [159]. Furthermore, pharmacological inhibition of mTORC1 results in the activation of GCN2 and the phosphorylation of eIF-2α [160]. The GCN2/eIF-2α signaling pathway is responsible for facilitating the process of autophagy in bovine mammary epithelial cells (BMECs) in response to IFN-γ [161]. However, the supplementation of Arg can reduce the occurrence of IFN-γ-induced autophagy in BMECs by inhibiting the GCN2/eIF2α pathway [162]. The induction of autophagy caused by acute AA deprivation suppresses intestinal inflammation by relying on the activity of GCN2 [160]. These findings suggest that GCN2-driven autophagy triggered by AA starvation is an important negative regulator of inflammatory stress and organ damage, which may be a therapeutic target for atherosclerosis.

### 5.3. Amino Acids Regulate Autophagy via AMPK Pathway

AMP-activated protein kinase (AMPK) is mostly recognized as a detector of intracellular energy homeostasis. Over the past decade, researchers suggested that energetic stress caused by prolonged glucose deprivation or mitochondrial toxins (i.e., an increase in the AMP or ADP/ATP ratio) inhibits mTORC1 signaling by AMPK phosphorylation and ULK1 activation, leading to autophagy activation [163,164,165]. A study by Ghislat et al. showed that, in mouse fibroblasts (NIH3T3 cells), withdrawal of AAs activates AMPK via Ca^2+^/calmodulin-dependent kinase kinase-β (CaMKK-β) and induces autophagy by activating ULK1 [166]. Furthermore, the effect of AA starvation on autophagy was weakened when AMPK was silenced.

Recently, two studies have challenged the prevailing view that AMPK promotes autophagy and instead proposed that AMPK activation inhibits the initiation and progression under AA deprivation [167,168]. Park et al. found that ULK1 is phosphorylated when AMPK is activated, thereby stabilizing the AMPK-ULK1 interaction and preventing the activation of ULK1 during AA starvation to prevent abrupt induction of autophagy [168]. Kazyken et al. found that activation of AMPK inhibited autophagy induced by AA withdrawal or mTORC1 inhibition. During AA deprivation, AMPK contributes to the reactivation of mTORC1 signaling, which supports essential cellular processes to promote cell survival [167]. Further research is needed to delineate the mechanisms by which AMPK promotes mTORC1 signaling during AA deficiency.

## 6. Conclusions

Many lines of evidence support the interplay of autophagy with AA availability and metabolism in atherosclerosis. Recent findings have provided new insights into the pathogenesis of atherosclerosis, which is not just a disease characterized by lipid accumulation in the blood vessels but rather a chronic inflammatory disease. This evolving understanding points to the importance of immune cells and their interaction with non-immune cells in the progression of atherosclerosis. The activation of effective autophagy in cell types involved in atherogenesis plays a protective role in reducing atherosclerosis development and maintaining plaque stability. While the canonical biological role of AAs in protein synthesis is clear, they can affect intracellular metabolism, autophagy, and consequently atherogenesis in many different ways. While certain AAs, when in excess, can inhibit autophagy and exacerbate atherosclerosis by promoting foam cell formation, oxidative stress, and plaque instability, restricting AA intake may have the opposite effect by enhancing autophagy and stabilizing plaques. The detailed mechanisms of how AAs can modulate the cell-specific processes of autophagy in atherosclerotic cells remain to be elucidated. Further experimental studies using either systemic supplementation or cell-specific AA delivery via nanoparticles are needed to decipher the intricate relationship between AAs and autophagy, particularly in vascular pathophysiology. This new evidence could propel the development of therapeutic strategies to leverage these mechanisms as druggable targets to improve vascular health and combat atherosclerotic CVD, thus reducing its extremely high burden of related morbidity and mortality.

## Figures and Tables

**Figure 1 biomolecules-14-01557-f001:**
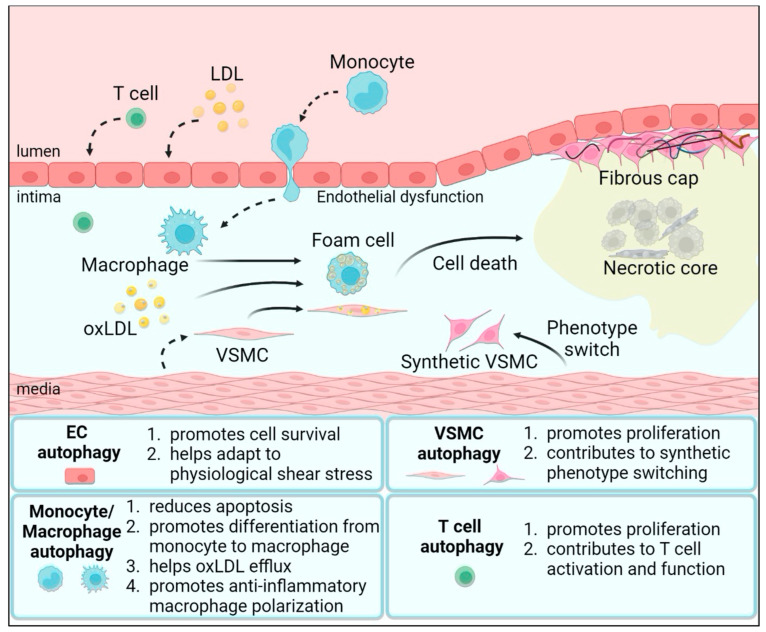
Pathogenesis of atherosclerosis and cell-specific contribution of autophagy in atherogenesis. In large- or medium-sized arteries, hemodynamic forces generate low or oscillating shear stress that causes endothelial dysfunction. This triggers LDL retention and oxidation in the intima. oxLDL stimulates endothelial cells to express adhesion molecules, leading to monocyte, T cell, and mast cell transmigration into the intima, where monocytes differentiate into pro-atherogenic macrophages. Along with VSMCs, monocytes engulf oxLDL and generate foam cells, indicating the formation of early fatty steak lesions. In the intima, T cells secrete cytokines (e.g., TNF-β, IFN-γ, and lymphotoxin) that induce VSMC migration from media to intima. Synthetic VSMCs proliferate and synthesize extracellular matrix proteins, which contribute to the formation of a fibrous cap over the lipid-rich core. Foam cells undergo apoptosis and release lipids and cell debris, leading to the formation of a necrotic core. The necrotic core and the fibrous cap are the hallmarks of advanced atherosclerosis. During atherogenesis, autophagy regulates many different aspects of immune and non-immune cells involved in the development of atherosclerosis. LDL—low-density lipoprotein; oxLDL—oxidized LDL; VSMC—vascular smooth muscle cell; EC—endothelial cell.

**Figure 2 biomolecules-14-01557-f002:**
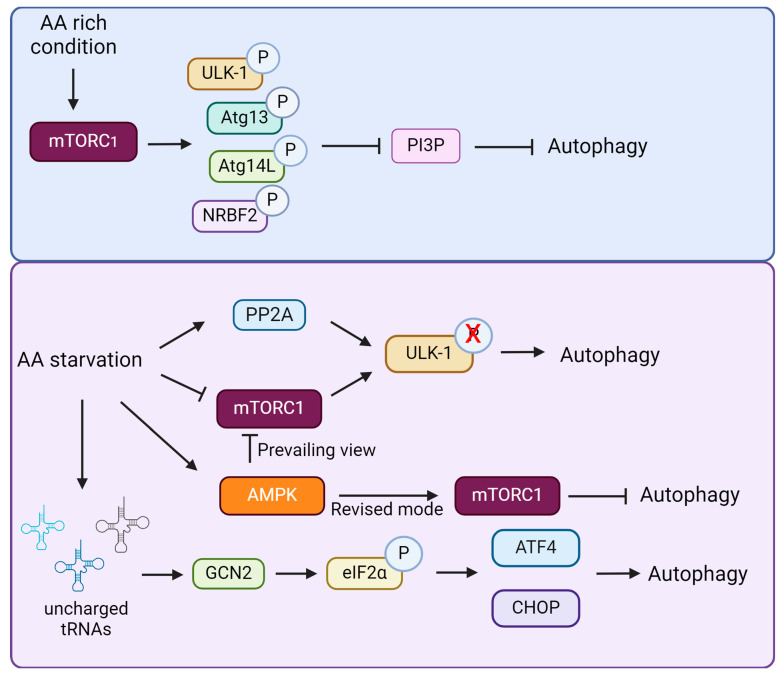
Regulation of autophagy by amino acids. When AAs are abundant, mTORC1 is activated and inhibits the production of PIP3 and the initiation of autophagy by phosphorylating ULK1, Atg13, Atg14L, and NRBF2. In contrast, AA starvation is activated by inhibition of mTORC1 or activation of PP2A leading to dephosphorylation of ULK1, which ultimately induces autophagy. In addition, AA starvation also affects autophagy by activating AMPK. The prevailing view is that AMPK activation inhibits mTORC1 and promotes autophagy. The recently revised view suggests that AMPK activation leads to mTORC1 activation and inhibits autophagy. Furthermore, the AA sensor GCN2 can affect autophagy at the transcriptional level. During AA deficiency, uncharged tRNA accumulates at the A site of the ribosome and interacts with GCN2, leading to eIF2α phosphorylation. Phosphorylation of eIF2α increases the expression of ATF4 and CHOP, which in turn promotes autophagy by facilitating the transcription of autophagy-associated proteins. AA—amino acid.

**Figure 3 biomolecules-14-01557-f003:**
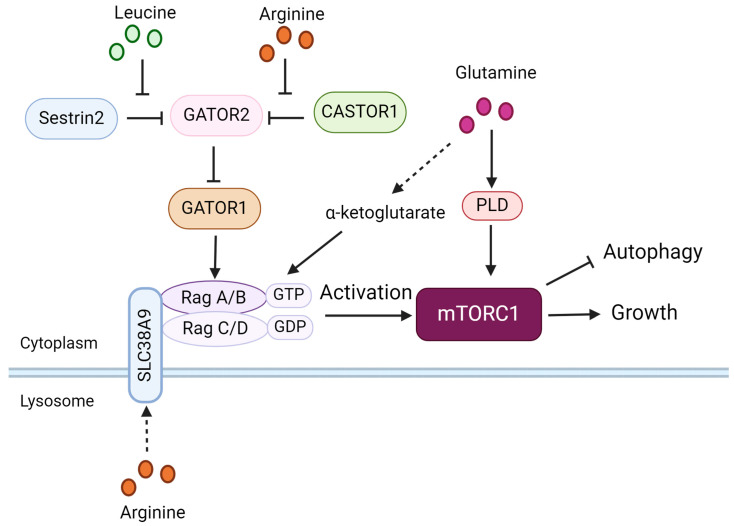
Amino-acid-dependent control of mTORC1 signaling. Sestrin2 acts as a Leu sensor and separates from GATOR2 when Leu levels increase, which results in increased activity of GATOR2 and subsequently suppression of GATOR1, followed by the activation of Rag A/B and mTORC1. Arg binds to CASTOR1 and drives CASTOR1 to separate from GATOR2, resulting in the activation of mTORC1 signaling. SLC38A9, an Arg sensor, interacts with the Ragulator-Rag GTPase complex and stimulates the activity of mTORC1. In addition to Gln’s activation of mTORC1 through activation of PLD, its metabolic intermediate α-ketoglutarate also stimulates Rag GTPase activation to turn on mTORC1 signaling. Arg—arginine; Leu—leucine; Gln—glutamine.

## Data Availability

No new data were created or analyzed in this study.

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
