# Peer review of "Amino Acid Metabolism and Autophagy in Atherosclerotic Cardiovascular Disease"

_biomolecules, 2024, doi:10.3390/biom14121557_

Round 1
Reviewer 1 Report
Comments and Suggestions for Authors
The review deals with autophagy, a mechanism important for the homeostasis of immune cells. Autophagy is important in the development of atherosclerosis and is regulated by the availability of amino acids, mainly leucine, arginine, and glutamine. The review summarizes the knowledge about the relationship between autophagy and atherosclerosis and the role of these amino acids in these processes. The paper is comprehensible, clear, and logically structured, it refers to recent literature. I recommend accepting the paper for publication in the journal Biomolecules.
Author Response
Reviewer 1
We thank the Reviewer for carefully evaluating and appreciating our work.
Reviewer 2 Report
Comments and Suggestions for Authors
The authors examine cardiovascular disease and the influence of L-leucine, L-arginine, L-glutamine, and autophagy during atherosclerosis development. The rationale is based in part on autophagy being crucial during the development of atherosclerosis. The primary goals of the review are to highlight how the selected amino acids contribute to autophagy and assess the potential of possible related therapeutic approaches.
Specific comments regarding sections of the review:
- Introduction (1) – The section is concise and well-written.
- Pathogenesis of atherosclerosis (2)—Like the introduction, this section is concise and well-written. Figure 1 is one of the best summarizing the pathogenesis of atherosclerosis and autophagy that I have seen.
- The section deals with specific amino acids(3) – Several reviews have dealt with amino and atherosclerosis. The paper “Atherosclerosis Linked to Aberrant Amino Acid Metabolism and Immunosuppressive Amino Acid Catabolizing Enzymes” comes to mind (https://pmc.ncbi.nlm.nih.gov/articles/PMC7549398/pdf/fimmu-11-551758.pdf). The authors should give more emphasis on the controversies, particularly given the focus on how the amino acids discussed in this review were chosen to have potential therapeutically.
- Section dealing with specific amino acids (4) – Given the vast amount of previous work focusing on amino acids/proteins and atherosclerosis – Why were L-leucine, L-arginine, and L-glutamine chosen? Alternatively, why were others excluded (e.g., owing to the large amount of available information dealing with branched-chain amino acids and atherosclerosis)? That said, the sections dealing with the roles of various signaling pathways were clear and convincing. As with the previous figures, the associated Figures for section 4 were excellent.
- Conclusion Section—This section provides a good summary of the material presented. The responses to the questions raised in sections 1-4 may, however, lead to more definitive statements than ‘more work needs to be done,’ e.g., suggestions of specific areas where there is the best likelihood of advance.
Author Response
Reviewer 2
We thank the Reviewer for his/her suggestion to further improve our manuscript.
1. The section deals with specific amino acids(3) – Several reviews have dealt with amino and atherosclerosis. The paper “Atherosclerosis Linked to Aberrant Amino Acid Metabolism and Immunosuppressive Amino Acid Catabolizing Enzymes” comes to mind (https://pmc.ncbi.nlm.nih.gov/articles/PMC7549398/pdf/fimmu-11-551758.pdf). The authors should give more emphasis on the controversies, particularly given the focus on how the amino acids discussed in this review were chosen to have potential therapeutically.
We appreciate the Reviewer’s comment and apologize for not making clear how we selected the amino acids. In contrast to the suggested paper “Atherosclerosis Linked to Aberrant Amino Acid Metabolism and Immunosuppressive Amino Acid Catabolizing Enzymes”, our article specifically highlights amino acids, which play crucial roles in autophagy, immune cells AND atherogenesis.
Based on Liu et al. the amino acids L-arginine, L-glutamine, L-leucine and L-methionine are functional in regulating autophagy. However, the sulfur-containing amino acid L-methionine is the precursor of homocysteine, which itself is implicated as a cause of occlusive vascular disease. Such enormous metabolic complexity makes it hard to decipher and evaluate the specific effects of L-methionine and is therefore beyond the scope of our present review.
Nevertheless, we fully agree with the Reviewer, that BCAAs play a pivotal role in the field of atherosclerosis. However, since we strongly believe, that specific targeting is the way to go to combat atherosclerosis and to keep the necessary immune system intact, we carefully decided to only include L-leucine as one BCAA and not the combination of all BCAAs.
In the revised version of our manuscript, we have now specified the selection as follows:
“To date, the AAs Arg, Leu and Gln have been reported to functionally regulate autophagy [54] and play a vital role in the progression of atherosclerosis. In this review, we therefore focus on the emerging roles of Arg, its derivative homoarginine (hArg), Gln, and Leu and their metabolisms in the key characteristics of atherosclerosis.” ll 220-224
2. Section dealing with specific amino acids (4) – Given the vast amount of previous work focusing on amino acids/proteins and atherosclerosis – Why were L-leucine, L-arginine, and L-glutamine chosen? Alternatively, why were others excluded (e.g., owing to the large amount of available information dealing with branched-chain amino acids and atherosclerosis)? That said, the sections dealing with the roles of various signaling pathways were clear and convincing. As with the previous figures, the associated Figures for section 4 were excellent.
Many thanks for appreciating our work. As outlined above, we agree, that it is important to clarify on why we only included specific amino acids. A respective explanation has been included in the revised version of our manuscript:
“To date, the AAs Arg, Leu and Gln have been reported to functionally regulate autophagy [54] and play a vital role in the progression of atherosclerosis. In this review, we therefore focus on the emerging roles of Arg, it’s derivative homoarginine (hArg), Gln, and Leu and their metabolisms in the key characteristics of atherosclerosis.” ll 220-224
3. Conclusion Section—This section provides a good summary of the material presented. The responses to the questions raised in sections 1-4 may, however, lead to more definitive statements than ‘more work needs to be done,’ e.g., suggestions of specific areas where there is the best likelihood of advance.
We thank the Reviewer for the suggestion and have now included more specific examples on how to precise future experimental work in order to target amino acid metabolism and autophagy to combat atherosclerosis:
“Further experimental studies using either systemic supplementation or cell-specific AA delivery via nanoparticles are needed to decipher the intricate relationship between AAs and autophagy, particularly in vascular pathophysiology. This new evidence could propel the development of therapeutic strategies to leverage these mechanisms as druggable targets to improve vascular health and combat atherosclerotic CVD, thus reducing its extremely high burden of related morbidity and mortality.” ll. 570-575